# Understandings of community participation and empowerment in primary health care in Emilia-Romagna, Italy: A qualitative interview study with practitioners and stakeholders

Daniela Rosalba Luisi[1,2], Kerstin Hämel[1]*

1 Department of Health Services Research and Nursing Science, School of Public Health, Bielefeld University, Bielefeld, Germany, 2 Department of Business Economics, Health and Social Care, University of Applied Sciences and Arts of Southern Switzerland (SUPSI), Manno, Switzerland

* kerstin.haemel@uni-bielefeld.de, kerstin.haemel@univie.ac.at

**Data Availability Statement:** Data cannot be shared publicly because of the potentially disclosive nature of entire interview transcripts.

## Abstract

Community participation (CP) and empowerment (CE) have long been viewed internationally as cornerstones of comprehensive primary health care (PHC). Accordingly, policies for new PHC models in Italy, such as the Community Health Centres called "Case della Salute" in 2006 and "Case della Comunità" in 2022, highlight the importance of implementing participatory processes with communities and creating opportunities for CE. This study's objective is to identify the understandings of CP and CE that emerge among practitioners and stakeholders who design participatory approaches in PHC practice and policy in the Emilia-Romagna region in Italy. Nineteen semistructured interviews were conducted with practitioners working on CP and CE processes in these Community Health Centres and with stakeholders involved in research on or the coordination of such processes in the context of these health centres. The data were analysed using qualitative content analysis in light of the following two questions, which emerged inductively from the data: (1) How to support CP and CE processes in practical doing (how do CP/CE)? (2) With which function or aim to support CP and CE (why do CP/CE)? This study shows that the participating practitioners and stakeholders exhibited various understandings of CP and CE in the context of PHC. Four main themes were identified: CP and CE as (a) a variety of forms of dialogue and cooperation, (b) tools for service development, (c) levers for empowerment, collectivism, and democracy and (d) stimuli for institutional change and a new level of professionalism. Moreover, the participants defined "the community" in different ways and often chose specific subgroups within the community to promote CP and CE processes. This study elucidates different perspectives on CP and CE and highlights the opportunities and obstacles for policymaking, research and practice that result from these understandings.

## Introduction

During recent changes in European health care systems, there has been increasing interest in the extension and reinforcement of primary health care (PHC) [1,2]. PHC is a comprehensive

Data are deposited on a secured network drive at Bielefeld University. They are available on reasonable requests from researchers who meet the criteria for access to confidential data. The data underlying the results presented in the study are available from the institutional representative at the Department of Nursing Science and Health Services Research. Please contact: data-access-dept.nursing@uni-bielefeld.de.

**Funding:** The author(s) received no specific funding for this work.

**Competing interests:** The authors have declared that no competing interests exist.

concept that refers to strengthening the health and well-being of individuals, communities and populations by integrating health promotion, prevention, treatment and rehabilitation as well as palliative care throughout the life span and in people's living environments [3]. Based on the values of social justice and equity, PHC addresses the social determinants of health (e.g. sociodemographic and sociopolitical factors, as well as circumstances in daily living) and fosters the empowerment and participation of individuals and communities [3–5].

Communities play an active role in health matters by developing, making decisions and taking actions that impact their living conditions [6]. Such processes should facilitate developing needs-based services, improving service quality and promoting health equity and social justice [7–9]. The task of strengthening community participation (CP) is closely related to the idea of "empowered communities" that develop skills and resources, build capacities and have space for reasoning and power sharing to ensure that social change can occur [6,10].

Many scholars have delineated typologies of CP and community empowerment (CE) in the context of PHC and public health [4,6,11–13]. Brunton et al. [11] suggested a conceptual framework to inform about the different perspectives and dimensions of community engagement interventions in public health. The framework points to two perspectives of participation of the community: the health services or '*utilitarian*' health system perspective, and the community engagement or '*social justice*' perspective. Both perspectives are distinguished by the extent to which CP approaches concern the community. From a 'utilitarian' health system perspective, CP approaches are focused on health interventions; CP and CE are 'tools' for developing health interventions and improving their effectiveness. When a 'social justice' perspective is taken, the focal point is the empowerment and engagement of the community to foster health, democracy and accountability in health and to face power imbalances between professionals and members of the community [11]. Similar to Brunton et al. [11], other scholars have recognised distinct directions of CP. Draper and colleagues [7] and Rifkin [12] described three perspectives on CP: In the medical approach, CP is directed by health professionals and aims at mobilising people to reduce illness and improve the environment. In the health services approach, professionals define needs and CP functions ways to mobilise community resources and engage people in health service delivery. In the community development approach, spaces for empowerment are created and community members decide over needs, as well as over the planning and managing of activities and resources. With this, the literature shows different theoretical perspectives of CP and CE and provides insights into the different actions, aims, and meanings attributed to CP and CE by those who promote CP and CE processes.

For our study, three dimensions of Brunton's and colleagues' [11] framework are of particular interest because they provide the foundation for organising CP and CE processes and are related to the understandings of CP and CE held by initiators of such approaches: the *definition of the community*, the *motivations or aims* the community is invited for and is interested in engaging, and the *actions taken*. According to Brunton et al. [11] to define the community it can be considered a community of interest or a community of geography, or, using a population view, the community can be defined in terms of a population with specific needs or socioeconomically disadvantaged group. The motivations or aims for developing and taking part in CP and CE approaches can differ for community members and professionals initiating them. Community members might be invited to participate, for example, for reasons related to ethics and democracy, better health and services, political alliances and resource leveraging. Community members' interest in engaging may be related to personal (wealth/health) or community gains, responsible citizenship or ideology. These two aspects are also supported by Cornwall [14] who suggests identifying "the community", those who are involved and those who are not, and clarifying the aims behind CP and CE approaches to better understand of the meanings attributed to CP and CE. Moreover, Brunton's and colleagues' [11] framework presents

various actions and processes that are relevant to CP interventions, such as the degree of collective decision-making, time for building relationships, and administrative and financial support.

CP and CE are characterised by a broad and diverse range of perspectives and dimensions. However, as described by various scholars, this is why these approaches are also difficult to grasp in theory and practice [7,14,15]. For this reason, it is thus important to analyse how stakeholders and practitioners conceptualise CP and CE, which determines how theory is applied in practice.

In Italy, since 2006, the Community Health Centre Model "Case della Salute" ("Houses of Health") has been propagated in the context of PHC to provide comprehensive, interprofessional and intersectoral community care [16–18]. However, as scholars have pointed out, due to its decentralisation, historically, there have been large regional disparities in access to and the provision of health care within the National Health Service in Italy [19–21]. This also holds true for PHC. At the beginning of introducing the Community Health Centre Model, the regions autonomously decided how to implement it. This led to differences in the organisation, in the service spectrum and in the nomenclature (e.g., in many regions, such as Emilia-Romagna, they were called Case della Salute; in others they were called, for example, Complex primary care units—Unità Complesse di Cure Primarie) [19]. While many Italian regions struggle with developing the health centres [19], only the Emilia-Romagna and Tuscany regions have managed to implement them widely [22].

With the upheaval caused by the COVID-19 pandemic, the importance of a strong PHC system has again moved to the foreground. In 2021, with the help of the "Next Generation EU" programme funded by the European Union, Italy decided to implement a *National Recovery and Resilience Plan* (*Piano Nazionale di Ripresa e Resilienza*) funded with two billion euros. One of its goals was to enforce the implementation and further development of the Community Health Centres [17]. To underscore their community orientation, the name of the health centres was changed to "Case della Comunità" ("Houses of the community") [21,23,24] in all regions.

These Community Health Centres are defined as easily accessible PHC organisations within the community that foster a new vision of health, enforce prevention and health promotion and integrate health and social services. They are managed by local health care agencies, divided into local health districts that manage and coordinate health services in a specific territory [19]. The standard services they offer include general practitioner and paediatrician services, home care and nursing, and social services. Depending on their organisation (as a hub or spoke model), they may also offer basic diagnostic services, family counselling services, vaccinations, screening programmes, mental health services, health promotion and prevention services and sports medicine services [19,23]. A particular feature of this primary health care centre model is that it also provides specialist outpatient services for high-prevalence diseases, e.g. from cardiologists, diabetologists, pneumologists, etc. [19,22,23]. A primary objective is the mandatory development of participatory approaches. As specified by the interministerial decree of the 23rd of May 2022 [23], CP and coproduction through collaboration with citizens' and voluntary associations, the local community, patients and caregivers, are obligatory features of each Community Health Centre [23].

In Italy, the Emilia-Romagna region has often been taken as a prime example for its organisation of PHC [21,25] and recognised as a particularly successful case at the national level [26] due to its early efforts to implement the health centres and support CP and CE. At the beginning of 2024, 128 Community Health Centres were distributed throughout Emilia-Romagna [27]. Other regions did not report similar progress [22]. Moreover, the Emilia-Romagna region has a long history of involving third sector organisations as representatives of the

community, and of participatory approaches in the regional and local health care system [28–30]. Moreover, similar to Tuscany, but different from other Italian regions, shared decision-making between the region and local health care agencies is very strong [20]. Currently, Emilia-Romagna's health centres are a stable part of its PHC service network [20]. Moreover, significant progress has been made to promote participatory and empowering approaches. For example, regional health authorities conduct and finance projects to generate knowledge and develop competences pertaining to CP among practitioners [31]. In these projects, experts who are specialised in participatory processes accompany and train practitioners from health centres and other local organisations. Such a supportive environment, and the long-standing experience and successful implementation of the Community Health Centres renders Emilia-Romagna nationally and internationally an interesting case for analysing how practitioners and stakeholders understand and ultimately shape CP and CE.

## Aim of the study

To our knowledge, only few studies have focused so far on the understanding and meaning of CP and CE for actors, such as practitioners and stakeholders, who design participatory approaches in both policy and practice in the context of PHC. Moreover, there is a lack of literature on this topic from high-income countries, including Italy. This study aims to address this knowledge gap by identifying the understandings of CP and CE that are adopted by involved practitioners and stakeholders in the context of PHC in the Emilia-Romagna region. The findings of our study contribute to our knowledge regarding how CP and CE are conceptualised in PHC settings that are promoting these approaches. This study also informs policies and practices related to CP and CE in health centres by supporting conclusions regarding the opportunities and obstacles that are associated with different meanings and connotations of what CP and CE should be.

## Methods

### Study design

A qualitative study design was used to analyse the understandings of CP and CE by drawing on good practices from the Emilia-Romagna region. A qualitative approach allows us to explore perspectives and experiences of actors in-depth and in detail in specific and complex contexts, so to understand nuances of a phenomenon [32,33], i.e. community participation and empowerment. The focus lies on the identification of rich and dense categories and codes without weighting one more than another. We conducted semistructured interviews with practitioners at the regional and local levels and with stakeholders both from the Emilia-Romagna region and nationally who had specific process- and context-related knowledge [34] of how CP and CE are understood in the context of the Community Health Centres. We analysed these interviews using qualitative content analysis [35] with the aim of identifying themes that could improve our understanding.

### Sampling and field access

Participants in the study were selected sequentially using purposive and snowball sampling following Patton [36]. We first looked for stakeholders from the PHC field who also had expertise in developing CP and CE in health centres in Italy nationally, regionally, and locally. We identified the stakeholders with the help of literature analysis, through PHC networks, national and regional governmental websites, and documents. Stakeholders were contacted and invited for an interview to provide us detailed information on CP and CE in the context of the

Community Health Centres in Emilia-Romagna. We also asked them to tell us about good-practice examples of CP and CE in the health centres and refer us to them.

This study defined good practices as projects/activities that put a strong focus on the development of CP and CE in the context of the health centres. We identified these good practice examples by desk search and by stakeholder recommendations.

For the good practice projects/activities, we identified practitioners involved in promoting CP and CE approaches. Practitioners were required to have practical or theoretical, process- and/or context-related knowledge of approaches to CP and CE in the context of health centres; they could be from interprofessional and intersectoral teams that support CP and CE activities, for example, general practitioners, nurses, and specialist doctors, as well as pedagogues and psychologists etc.

During the practitioner interviews, further good practice activities from other health centres were recommended to us. Sometimes, interviewed stakeholders or practitioners gave us the names of specific contact persons involved in CP and CE in a specific Community Health Centre; and sometimes, they told us about a good-practice activity in a health centre. In that case, we would contact the local health care agency in charge of that centre and request a referral to the contact person responsible for the development of CP and CE.

Notably, health centres can vary from each other as they adapt their services and their CP and CE strategies to local contexts, geography, and populations. Likewise, local health care agencies that manage health centres can differ in their organisation, services, focus and resources provided [37], as well as with respect to CP and CE processes. Our focus on good practices made it necessary to include Community Health Centres from all over the regional territory to build a robust sample. This is justified also by the fact that not all health centres that are managed under the same local health care agency are fostering CP and CE equally. Therefore, we selected a sample of good-practice examples from different organisational and local contexts so to reveal various developments at the local level, as well as similarities and differences that go beyond local contexts. This greater diversification of cases and contexts was purposively chosen to increase the validity of our findings on a broad empirical basis. At the same time, it can provide insights into the possible influences of organisational and local contexts on understanding and action in practice.

When choosing good-practice examples, we looked for Community Health Centres in urban and rural areas, different provinces of the region, and under different local health care agencies to increase the variety of information-rich cases and the validity of our results. We searched for health centres supporting CP and CE processes until no more health centres could be found.

Finally, we identified interview partners giving us information about 14 Community Health Centres from six out of nine different provinces of the Emilia-Romagna region. In the remaining three provinces, we either could not identify good-practice examples or practitioners were unwilling to participate in the study. Sometimes, to obtain a clearer picture of the process, we interviewed more than one practitioner involved in the same CP and CE process at one health centre.

Recruitment took place from the 2nd of October 2020 until the 31st of May 2021. Study participants were contacted via e-mail or telephone and provided with written information about the study before the interview. The recruitment of further potential study participants ceased when the information we collected allowed us to delineate themes comprehensively and when the information given by the interview partners started to be repeated (data saturation).

Table 1 gives an overview of the study sample (n = 19), which included six stakeholders and 13 practitioners. Two of the practitioners also reported their experiences as citizens who participated in CP projects. One stakeholder had previous practical experience with CP/CE

**Table 1. Characteristics of the sample (N = 19).**

| Participant characteristics | Number of participants |
|---|---|
| **Participants, total** | **19** |
| **Practitioners** | **13** |
| Community Health Centre staff<br>involved in the coordination and/or creation of community participation and/or community empowerment processes in a Community Health Centre | 5 |
| Cooperating professionals<br>from a local health care agency or social services who were involved in the coordination and/or creation of community participation and/or community empowerment processes in a Community Health Centre | 8 |
| **Stakeholders** | **6** |
| Representatives of third sectors at the national level | 2 |
| Researcher and project coordinators<br>of community participation and/or community empowerment processes in Community Health Centres/primary health care at the national or regional level | 2 |
| Managers of local primary health care/ Community Health Centres | 2 |
| **Gender** | |
| Male | 9 |
| Female | 10 |
| **Professional background** | |
| Nurse | 3 |
| Physician | 8 |
| General practitioner (specialisation in general medicine) | 3 |
| Other (e.g. specialisation in hygiene and prevention) | 5 |
| Pedagogue | 2 |
| Other (e.g. studies in health management, psychology, economics) | 6 |
| **Focus of expertise** | |
| Local level | 14 |
| Regional level | 2 |
| National level | 3 |

approaches in the context of PHC. In total, these interview partners provided us with detailed descriptions of CP and CE approaches in 14 Community Health Centres.

## Ethics

Ethical approval for the study was granted on 29 of May 2020 by the Ethics Committee of Bielefeld University according to the guidelines of the German Association of Psychology (Deutsche Gesellschaft für Psychologie; DGPs) which correspond to the guidelines of the American Psychological Association (APA) (Application No. EUB-2020-091). All study participants gave written consent after receiving participant information that provided them with all the relevant information regarding the study and its goals as well as data protection and privacy.

## Inclusivity in global research

Additional information regarding the ethical, cultural, and scientific considerations specific to inclusivity in global research is included in the Supporting Information (S1 Checklist).

## Data collection

An interview guide [38,39] (S1 File) was developed with a focus on the process and context knowledge [34] of the participants regarding to the following topics: (a) past, recent and future developments in the Community Health Centres, (b) understandings of CP and CE, (c) general evaluations of CP and CE in the Community Health Centres, (d) the transfer of CP and CE policies into practice and possible ways of improving this transfer, (e) examples of CP and CE approaches, (f) hindering and supporting factors pertaining to participatory and empowering processes and (g) CP in times of physical distancing. The only difference between the interview guidelines for the practitioners and those for the stakeholders pertained to the question concerning examples of CP and CE approaches in practice. In this context, stakeholders were asked whether they had experienced or knew about approaches pertaining to CP and CE and, if so, whether they could describe them. This research is part of the larger PhD project of the first author. For the present study, we analysed data regarding participants' understandings of CP and CE and their general evaluations in the context of the Community Health Centres in the Emilia-Romagna region.

The nineteen semistructured interviews, which lasted 1–3 hours each, were conducted between October 2020 and May 2021 by the first author, who holds a bachelor's degree in occupational therapy and a master's degree in public health and who has previous experience conducting several interview studies throughout her academic and professional career. Originally, face-to-face interviews were planned to ensure a natural atmosphere [40]. However, northern Italy, including Emilia-Romagna, was particularly impacted by the spread of COVID-19. As the first author resides in Switzerland, at some points during the pandemic, she was not able to enter Italy. Therefore, all the interviews were conducted online via Zoom. Online interviews are known to have limitations such as the need for technical instruments (e. g. computer and internet access) and the risk of technical problems, such as an internet connection failure, which could disrupt the interview [39,41]. Furthermore, some scholars have noted that the quantity and "richness" of data generated by online interviews may be lower; however, a recent comparative study indicates that visual online data collection modalities produce an amount of in-depth data that is comparable to that of in-person interviews [41]. As it was not possible to conduct the interviews in-person, the visual online format served a good alternative. It also provides a high degree of flexibility in scheduling the interviews [39,41], which was particularly relevant in this study for the health care practitioners. Before conducting the interviews, the first author mitigated the risks by checking all the technical aspects in advance (e. g. internet connection, audio) and making sure that the interview partners could hear and see each other properly. Furthermore, she tried to establish a welcoming and comfortable conversational atmosphere by, for example, thanking for the interview, introducing herself and the topic, asking questions about any concerns or doubts (also about technical issues), demonstrating a positive verbal and nonverbal attitude, and ensuring a calm environment free from distractions or noise. She also took some brief notes during the interview so that she could quickly catch up on the content in case of interruptions.

## Data analysis

The interviews were audio recorded, fully transcribed verbatim using the computer tool 'easy-transcript' (e-werkzeug) and anonymised. Word documents and MAXQDA 12 software (VERBI GmbH) were used to facilitate the coding of the transcripts. For the data analysis, we performed qualitative content analysis based on the work of Kuckartz [35,42,43]. First, the first author deductively developed a coding system featuring different themes based on preliminary ideas and questions drawn from the interview guideline. In addition, when the material was

**Table 2. Themes related to participants' understandings of CP and CE.**

| Community participation and empowerment as . . . |
| --- |
| (a) a variety of forms of dialogue and cooperation |
| (b) tools for service development |
| (c) levers for empowerment, collectivism, and democracy |
| (d) stimuli for institutional change and a new professionalism |

examined in a first reading by both authors, new ideas and codes were derived inductively from the data and subsequently discussed. Two different perspectives emerged regarding two specific aspects:

1. How to support CP and CE processes in practical doing (how do CP/CE)?

2. With which function or aim to support CP and CE (why do CP/CE)?

Emerging themes were discussed repeatedly by the two authors and elaborated until consensus was reached. Through a triangulation process and discussion between both authors, coding rules for the themes and subthemes were discussed and documented to ensure consistent data coding and increase the confirmability of the results. Among other topics, the discussion concerned the coding tree in terms of its comprehensive characterisation of how the interview participants understood and conceptualised CP and CE in the context of the Community Health Centres in Emilia-Romagna. Table 2 shows the four themes observed.

The data were coded in their entirety in accordance with these themes by the first author, who drafted a detailed interpretation, which was discussed several times with the second author until a consensus regarding the interpretation was reached. The second author is the supervisor of the PhD project of the first author, who is located in Germany, holds a doctorate in social sciences and has comprehensive experience in conducting qualitative research on health services in foreign countries.

We used different criteria to support the trustworthiness of our qualitative study [32,44,45]. We engaged in researcher triangulation to reduce possible bias in the data collection, analysis and interpretation. A reflective approach was used during the whole research process: Notes and memos reflecting ideas, preconceptions and presumptions of the researchers were taken. In addition, the triangulation of resources was achieved by choosing participants with different professional backgrounds who could provide in-depth information about CP and CE in the Community Health Centres in Emilia-Romagna from national, regional, and local perspectives. These are important criteria for ensuring the credibility of our study. Moreover, the clear articulation of our sample strategy supports the transferability of the findings. We provided a detailed description of the research methods and process to ensure dependability of our results.

## Results

The interview participants indicated that for the development of CP, "experiences are a bit patchy" (Stakeholder-5_par.5) and that there are very few health centres that truly try to implement CP and CE. In the absence of general guidelines for promoting CP and CE, the approaches "have been left a bit to the local will and intuition" (Stakeholder-5_par.5). Whether or not CP and CE are fostered "depends a lot on the directions of the local health care agencies" (Practitioner-1-health management_par.52), including each local health care agency's willingness to invest resources in CP and CE.

**Table 3. Interview participants' understandings of "the community".**

| "The community" is composed of... | Examples |
| --- | --- |
| (1) users of the Community Health Centre, community members living in the respective community, and/or their associates | Community members/citizens, volunteers, users' and citizens' organisations, sport clubs, churches |
| (2) professionals from local services | Professionals working in social care services, schools |
| (3) representatives from the municipality and the local health care agency | Practitioners working for the local health care agency, practitioners working for the municipality |
| (4) shop personnel | Personnel from barber shops, bakeries, pharmacies, and supermarkets |

In addition, the promotion of such processes is, according to our analysis, strongly influenced by the ideas and values of the management and the practitioners in the health centres, as well as by those of other community actors. The development of CP and CE depends on the local culture and perspective of the people working in a single health centre:

"[. . .] there are other Case della Salute like the one in [village]; this one also has a community vocation with family doctors who have this thinking, but especially with a very strong nursing component that was born with this culture. So, it depends on the various territories. [This is] what we tried to do, when in the local health care agency, we started to build right even organisationally the Case della Salute, because the cultural inspiration was there even before"

(Practitioner-9-physician_par.41).

The interview partners highlight a supportive local culture for participatory activities and the advantage of drawing on preexisting networks as resources to foster CP.

"Because we know that there are regions or territories that already have a bit of this mission, of this nature. That is, for example, in our reality, our administrators, our territories, were already very accustomed to dialogue with citizenship, with the communities. We also have a rich associative network of nonprofit or third sector associations, with whom we have codesign tables, dialogue tables; here, so there is this custom. But we know that not all territories are the same [. . .]

(Practitioner-3-pedagogue_par.8-9).

When the interview participants refer to those networks as "the community", they describe a broad spectrum of persons and institutions. Table 3 offers an overview of their understandings of "the community".

## Community participation as a variety of forms of dialogue and cooperation

The interview participants relate their understanding of CP to a variety of forms of dialogue and cooperation between health practitioners and the community. In this respect, dialogue entails an exploration of the viewpoints and needs of the community. To accomplish this task, they see the need for "fostering opportunities for listening to the community" (Stakeholder-7 _par.14). Listening (*ascolto*) is viewed as an important prerequisite for guiding the work of practitioners and their ability to respond appropriately to community needs. Practitioners support such dialogue, for example, by informing community members about health

issues and healthy lifestyles. Simultaneously, practitioners provide spaces to which participants are invited to relate their own experiences with and viewpoints on health-related issues from the perspective of their lifeworld. In this way, the limitation of unidirectional information flows from practitioners to healthcare users can be overcome. Moreover, by engaging in consultation and joint discussion meetings (e.g., public meetings, world cafés, and focus groups), practitioners try to understand people's needs as well as their ideas and opinions concerning health and social issues. This approach is viewed as a foundation for community work:

"[. . .] to do community participation and create dialogical spaces where I allow people, I create the right conditions, so that people can intervene and have their say on things that are always complex. If not, we see only our point of view. We need to understand the points of view of all the subjects who come into that policy, into that object. For me, this is participation, and therefore knowing how to set up dialogical spaces"

(Practitioner-1-health management_par.42).

Interview participants also describe forms of cooperation with the community that go beyond the level of mere dialogue with and consultation of the community. For them, involving community members and other community representatives entails setting priorities and goals jointly, reaching agreements among the relevant actors and coplanning specific activities.

"I think that community participation has to go [along] [. . .] with small but clear and concrete things on which to ask citizens for cooperation. [. . .]. That is, not necessarily the doing, but just the development of ideas. That is precisely also the sharing of ideas or helping all the stakeholders in the redesign, the codesign"

(Practitioner-3-pedagogue_par.13).

To foster such cooperation, it is crucial to work both in and with formal and informal networks. According to the interview partners, formal networks among practitioners, professionals from local services, community representatives and members are crucial. Additionally, they highlight the particular significance of informal networks for fostering discussion (e.g., of different needs) and collaboration both with and between different subgroups of the community, e.g., pupils, adolescents, seniors, and caregivers of people with dementia.

"I belong to this community; you belong to another; do you reflect yourself in it? Alright, let us work to produce the meeting between these communities and build the network that circulates. . . more information and activities: this is community participation"

(Practitioner-15-physician_par.65).

The interview participants believe that the process of approaching and fostering such networks is an important strategy for activating existing community resources using a low threshold approach. Moreover, working with informal networks can help health practitioners identify difficult-to-reach (sub)groups in the community, for example, isolated older people, single mothers, or immigrants, and to offer them a voice in participatory processes.

## Community participation as a tool for service development

Interview participants who endorse the idea of CP as a proper tool for service development believe that involving communities makes the health centres more accessible, responsive, and valuable to the community. They emphasise the strategic use of CP to promote service improvement. According to this understanding of CP, users of health centres, citizens and their associations are involved in contributing to health service development through their own resources, e.g., volunteering. Promoting participatory processes is understood in this context as a means to develop community resources that can help 'optimise' the work of health centres and related services as well as to coproduce new services.

> "We [the practitioners associated with the Community Health Centre] recruited several volunteers from the community who joined the association [name of association] to contribute to a series of initiatives to support the family members of hospice patients"
>
> (Practitioner-12-nurse_par.106).

Another example is the incorporation of volunteers into the welcome hall of the health centre. These volunteers welcome and direct users to the appropriate services with the goal of making service and care paths smoother and clearer for users.

Moreover, the interview participants praise the fact that specific community members are engaged as low threshold mediators (to some degree) who can promote the connection between the health centre and the community as a whole. For example, shop personnel from barber shops and bakeries are asked by the practitioners of the Community Health Centres and the municipality to be sensitive and attentive to situations of vulnerability of their clients, to inform the clients about the services and, if so desired, to guide them towards the health centre or social services. Simultaneously, these intermediaries are viewed as being able to keep an eye on the community and thus as able to inform the practitioners regarding the community health and social problems that they perceive to ensure that the health centre can respond by providing adequate services. In this context, the interview participants refer to problems such as loneliness and poverty, drug abuse, and difficulties accessing maternal services for immigrant mothers.

Additionally, the involvement of community members can be used to assess the quality and appropriateness of health centres and their services from 'outside' and to obtain insights into possible means of service improvement.

> "They [the citizens] had the opportunity to have a direct discussion with the city administration that was mediated by the Casa della Salute, [. . .] and to also have a direct discussion on the organisation of the services of the Casa della Salute. Therefore, indeed, the health agency changed some decisions, some policies, based on the discussion with the group of involved citizens"
>
> (Practitioner-14-physician_par.9).

Additionally, CP, understood as the involvement of the community in service development, is also an important tool for defining the Community Health Centre itself. The health centre, as a local health organisation, is still little known about by the population and professionals from local services. It is often unclear what its roles and tasks are and how it differs from general practitioner practices or outpatient polyclinics. In this regard, the interview participants emphasise that the participation of the community in the context of the health centres can

help create awareness of what distinguishes it and the services that it should offer–not only on the part of the wider community, but also among practitioners in the health centre themselves.

## Community participation as a lever for empowerment, collectivism, and democracy

The third identified theme is the understanding of CP as a path towards the strengthening of social movements and the development of the community. In this respect, participatory processes are viewed as a lever that can foster people's ability to exercise their democratic rights, develop the capacity of community members, e.g., to engage in discussion and advocate for themselves, establish a sense of collectivism and help people gain influence over their own resources. The interview participants who highlight these aspects describe "the community" in terms of citizens and local (voluntary and citizen-based) associations. Frequently, they focus on community subgroups and disadvantaged groups, while they focus less often on cooperating institutions and professionals from local services. Accordingly, CP serves the goal of promoting broader social developments within the community that are not necessarily or not directly related to a specific health or social services outcome. However, the interview partners believe that a responsible, capable, and proactive community that features strong social ties positively influences the health and well-being of its members because they are able to acquire resources, e.g., for the purpose of mutual support.

In this regard, CP is viewed as a lever for collectivism; involving the community facilitates the development of "the awareness of also being a collective, of also being something that lives within group dimensions" (Stakeholder-17-physician_par.63). This vision of a collective requires the community members to find joint values and shared needs and to negotiate priorities and resources; this task can be accomplished through relationships among community members and the experience of reciprocal support. The interview participants believe that community members' development of such capacities helps them discover joint solutions to the challenge of acquiring health resources.

> "That is, the community itself finds within itself the strategies to overcome the health problem that the person has brought. That person listens and has a whole series of strategies that he can try to implement [. . .]. I am truly talking about dialogical spaces where people who do not feel that they have a pathological problem can come to talk, even simply about an anxiety, a worry [. . .] and when they have found a whole series of strategies and stories and have managed to calm this person down. . . So, for me, that is also doing community empowerment: creating those kinds of spaces"

> (Practitioner-1-health management_par.46).

As highlighted in the previous quotation, the idea of CP is closely related to CE. Described as "capacity-building" (Stakeholder-17-physician_par.55) or obtaining ability and "agency" (Practitioner-15-physician_par.34) within the community, CE is viewed as a requirement not only for mutual support but also for people's ability to self-organise, effectively claim their rights and participate in decision-making processes.

The interview partners, however, indicate that in most places, social ties are not so strong. Consequently, the promotion of networks, collectivism and spaces for empowerment as well as their sustainability over time require and must often be constantly accompanied by practitioners. In this context, they see it as insufficient to solely create networks based on single practitioners and community members who casually share the same ideas and values. Instead, they call for a systematic approach to establish regular (participatory) activities and investment from the side of the institutions to promote the establishment of social bonds.

"So, what we have been trying to do for years now is to create links; it is to create relation-ships that are not spontaneously built, not, let us say, germinated because there's a good feeling between whoever is in charge of that voluntary association with that professional manager of a Casa della Salute. However, relationships have to be built in an integrated sys-tem, in a real community building action [. . .]"

(Practitioner-9-physician_par.28).

In this regard, one interview partner highlights the importance of a bottom-up vision of CP. Practitioners and institutions can facilitate community networks, empowerment, and capacity-building, but ultimately, it must be the community that advocates for its own health concerns and mobilises its own resources.

"If we want to talk about participation, the point of observation must be reversed. It is not the institutions that build participation, but it is the community from within, is not it?"

(Stakeholder-4_par.129).

Interview participants who emphasise the idea of CP as a lever for empowerment indicate that practitioners who promote opportunities for CP and CE must be constantly aware of their "superior" position in terms of knowledge and expertise as well as their ability to exercise power. To respect the community members and their expertise regarding their own living situ-ations, the interview partners (once again) refer to the importance of "listening" to the com-munity and thus establishing the proper conditions for CP and CE.

Finally, they present their idea of CP as a means of exercising democratic rights and taking control within the health care system. As one interview participant argues, "community partic-ipation is simply another exercise in the production of de facto democracy that [the commu-nity exercises] on the territory" (Practitioner-15-physician_par.28). This task entails that all the different opinions and needs—even those that are contrasting or clashing—must be dis-cussed and considered.

## Community participation as a stimulus for institutional change and a new professionalism

The fourth theme identified focuses on CP as a catalyst for change in institutions and practi-tioners. Interview partners who highlight this catalyst function point to the "community of professionals" (e.g., practitioners in the health centre, the local health and social sector; manag-ers of the health centre and local health care agency) as a group that particularly benefits from involving and being involved in participatory processes. In this respect, participatory processes featuring users and community members are viewed as stimuli that can change practitioners' mindsets and working methods.

As one interview participant notes, participation represents the "ability of institutions not to be *for* but *with* the citizen. In addition, [professionals] give citizens recognisable spaces for them" (Stakeholder-4_par.44). Participatory processes can thus support practitioners and organisations in developing a more client- and community-oriented approach in the long term.

"So, to be able to respond to a health need, I need to go and ask people, to understand what their experience is. And so, for me, it was: I start doing it on individual projects, but it has to become a way of working of the health and social professionals. If not, we do not answer anymore; if not, we go one way, and they go the other"

(Practitioner-1-health management_par.22).

In this regard, participatory processes allow practitioners to recognise the potential that a "lay community" can offer. Moreover, direct involvement helps practitioners identify situations of vulnerability within the community of which they are usually not aware (e.g., during individual consultations) and that remain hidden.

"We [practitioners of the Community Health Centre] could succeed in achieving our objectives only if the community is already there. To have people who act as antennae for situations, and who feel themselves to be an integral part of the Casa della Salute and who can therefore stimulate access to subthreshold situations, to situations that we are not able to identify and highlight, and therefore to allow these barriers to be broken down"

(Practitioner-6-physician_par.44).

Consequently, the participation of users and community members in projects initiated by practitioners encourages practitioners to rethink and change their working methods and their attitudes towards their clients. If practitioners adopt the idea that the expertise of community members regarding their own lifeworld is as valuable as practitioners' clinical expertise and if these parties start to meet on a more equal basis, they can adopt a new perspective that promotes a more community-oriented approach.

"Because involving the community in a health project also means putting them at the same hierarchical level as any other clinical expert [. . .] The game is there. [. . .]. Obviously, everyone takes part in the coconstruction of a project for which they have expertise. Obviously, I, as a volunteer, do not tell the neurosurgeon how he has to do the surgery, but, for example, to the oncologist, I can tell him what happens when a patient goes home and has received a negative diagnosis in a certain way. So, that is a powerful bit of learning that changes and modifies the quality of care as well"

(Stakeholder-7_par.95).

However, the interview partners are also aware of the risk that practitioners may merely reduce CP and CE to a technique or technical act. Accordingly, they advocate for a professional attitude that is characterised by the idea of constant dialogue and exchange with the community as a matter of everyday routine.

## Discussion

This study explored the understandings of CP and CE expressed by practitioners and stakeholders who referred to good-practice examples to promote CP and CE in the context of Community Health Centres in Italy's Emilia-Romagna region. In line with Cornwall's [14] suggestions, practitioners' and stakeholders' descriptions of and narratives regarding CP and CE are first explicated by how they are supported in *practice*. They offer testimonial evidence of a variety of forms of dialogue and cooperation based on which their understandings of CP and CE evolve. Second, they refer to specific *aims* that they relate to the task of strengthening CP and CE.

Our analyses revealed three distinct aims or functions: CP and CE are viewed as a) tools for service development, b) levers for empowerment, collectivism, and democracy and c) stimuli for institutional change and a new professionalism. These three goals exhibit different, sometimes competing rationales. However, the stakeholders and practitioners interviewed in this

study often refer to more than one of these aims. This situation is in line with the results of a study conducted by Glimmerveen et al. [46], who showed that even within one single organisation or participatory process, different ideas about CP can coexist; this coexistence could lead to uncertainty among actors regarding who should be involved in this process and in what way. Some scholars argue that incoherent understandings of concepts such as CP are related to the variety of disciplines involved in this process, which use different terminology [47].

Moreover, in the case of the Community Health Centres, the lack of a shared understanding of CE could be explained by unclear definitions and scarce descriptions at the national and regional policy document levels [28]. These differences in common understandings of CP and CE could also be explained by the fact that different local contexts (e.g., management and commitment of the local health care agency, preexisting collaborations with the local community and third sector, readiness of the municipality, willingness of single health centres and its staff) use different approaches and are free to decide with which aims, how and with whom to engage in CP and CE processes, as also suggested by our findings. This is also confirmed by Glimmerveen et al. [46], who highlights that management practices and organisational context, which are specific to each local context, shape the way in which participatory processes are implemented, making it, therefore, difficult for PHC actors to develop a common idea within or even across regions.

To some extent, a joint, interprofessional and intersectoral perspective on the meaning of CP and CE is necessary as a basis for action. This is also supported by other scholars who advocate for a common frame of understanding [8,48] of practitioners, managers and policy-makers to truly develop CP and CE in PHC. Therefore, a joint perspective should be shared by all relevant local actors (in our example, e.g., the staff of the health centre, local health care agency, the municipality, social services, local associations, and members of community groups). Developing and communicating good-practice examples could offer an opportunity for relevant actors to think about and act for similar purposes.

Moreover, national and regional policy documents are needed to provide a common definition and description of what CP and CE mean in the context of PHC or specifically in the Community Health Centres in Italy, how these two approaches are interrelated, and with whom and how they can be facilitated in a comprehensive process. Policies and guidelines should support practice in reflecting on who they mean by "community" and finding consensus to create appropriate approaches. In this context, the heterogeneity of the communities and the specific local contexts that can influence CP and CE in practice should be considered [49]. Moreover, it should not be forgotten that a considerable part of the community (e.g., disadvantaged groups) might first require an approach that enables them to develop the capacity to participate genuinely in decision-making processes.

The interview participants relate participatory and empowering approaches in practice to the ideas of professionals listening to community needs, discussing opinions with the community, cooperating on specific topics, and helping create social bonds. Our results support the findings of much of the extant literature, which has also identified dialogue, consultation, and collaboration as important aspects of CP [50–52]. The relevance of supporting (informal) networks is particularly notable. Understanding CP in terms of creating social networks entails the attempt to establish relationships among people beyond a geographical area, i.e., relationships in which people *feel* that they belong to this network [53]. As the community is always heterogeneous and composed of members with different needs and values, meetings and exchanges can stimulate establishing such social bonds [53]. This ambitious task should be addressed jointly by various local actors (e.g., Community Health Centres, local health care agencies, the municipalities, social services, local associations, and community groups), and

practitioners and stakeholders should play the role of community partners throughout the process of creating meaningful relationships.

Our results reveal that the two aims of participatory approaches in the context of the Community Health centres are (1) the improvement of services and service development and (2) the promotion of empowerment, collectivism, and democracy. Similar rationales for CP have also been differentiated by Brunton et al. [11] and other scholars [7,12,13,46] as aims that often lie at two ends of a continuum of CP. The results from our study show, however, that in the interview partner's experiences and opinions, the two rationales can coexist, even within one organisation.

Drawing on Brunton's framework [11] and on the literature on CP and CE [7,12,13] with regard to the health service development perspective, scholars have criticised that main decision-making power remains in the hands of health professionals [7,11–13]. Our data do not show clear evidence to support this claim, especially when this goal exists alongside others. Rather, a sole perspective on service development entails the risk that wider effects on the community and, thus, the full potential of CP and CE may be overlooked. However, our findings resonate with the social justice perspective and its focus on supporting CE and communities' involvement in planning and decision-making processes [7,11–13]. The interview participants emphasis in their understanding of CP and CE issues related to (decision-making) power and collective capacity. Recent debates on participatory approaches have also focused on these aspects [53–56]. Among the many principles associated with the successful implementation of CP, addressing power imbalances seems to be one of the most important [54]. However, this task is challenging when power and control are not shared and when autonomy is not sufficiently attributed to the community by powerholders, e.g., practitioners and management [55]. This task also entails considering the processes that allow citizens from all/different subgroups to have access to decision-making processes with the goal of truly addressing the needs of the whole community. This goal also requires investments in capacity-building on the part of the powerless rather than relying on more powerful, prestigious groups within a community [54–56]. For all the actors involved in this process, being aware of the different rationales underlying CP and CE is important, as it has implications for their practice: their aims can guide their activities and the outcomes that they want to attain [13].

Another aim or function that the interview participants consider related to CP and CE is the stimulation of changes in the "community of professionals". According to the participants, practitioners' own participatory experience becomes a medium for professional learning about CP and CE and causes them not only to become aware of the value and importance of these approaches for their work but also to change the way in which they think about and work with the community. This finding is also confirmed by McEvoy et al. [47], who showed that involving practitioners and stakeholders in participatory projects helps them grasp the sense and meaning of CP and their own role more effectively. Thus, creating possibilities for more practitioners and stakeholders of Community Health Centres and PHC to become personally engaged in participatory processes could be a promising way of spreading ideas and knowledge pertaining to CP and CE within organisations.

In addition, our results show that participants in the study have distinct ideas of who "the community" is, of participatory processes in the context of PHC, and of the community subgroups that should be involved; for example, some participants identify community members with a general interest in health and social services or specific (disadvantaged) subgroups, while others refer to the "community of professionals". Some researchers highlighted the importance of clarifying who exactly constitutes the community, as this question influences the objectives of and planning for participatory approaches [13]. Moreover, a careful consideration of who should be involved helps us determine whether all relevant actors have been

involved. This is particularly relevant since the involvement of only specific target groups entails the risk that only a small part of the whole community may be represented. In this manner, actors could "invisibilize[d] members of the community who do not participate, silent or dissident voices of these actions" [53]. Practitioners and stakeholders must therefore be attentive to this aspect when planning to facilitate CP and CE.

Various lessons can be learnt from this study's findings. First, to our knowledge, few studies have analysed the understandings of actors involved in CP and CE in PHC in high-income countries. Our results provide first examples from Emilia-Romagna, allowing us to obtain an overview of the different understandings of CP and CE and their interrelations, which might also provide helpful insights for other PHC or Community Health Centres contexts. Second, although local contexts, single regions and different countries organise CP and CE in PHC differently, we found similar challenges related to a shared understanding of CP and CE and an uneven idea of who "the community" is and its roles in the process [47,48,57]. Clarifying CP and CE processes for local or regional PHC contexts by specifying exactly with which aim who is engaged in what can help direct specific actions and lower the risk of CP and CE remaining solely rhetorical [14]. Third, our findings align with scholarly critics about the importance of addressing power imbalances when fostering CP and CE processes [54–56]. As also confirmed by other scholars analysing CP in high-income countries, actors concerned with CP and CE should carefully decide how they intend to address power relations between involved actors [15]. Finally, this study adds evidence to the literature on the potential of CP and CE approaches experienced by practitioners themselves to improve their understandings and views on CP and CE [47]. This represents an interesting finding not only for initiators of CP and CE processes but also for managers who want to foster CP and CE in their organisations and who are concerned with their planning.

## Strengths and limitations of the study

We chose to collect data from different Community Health Centres in different provinces of the Emilia-Romagna region that were identified for this study as examples of CP and CE good practices. The findings reveal stakeholders' and practitioners' views on CP and CE, but not the understandings of local community members and users. User and citizen involvement in the study would have added additional perspectives on this topic, which should be explored in future research. However, as in the Community Health Centres in Emilia-Romagna, it is primarily the practitioners who are called to create spaces for CP and CE and to collaborate with the community and the third sector, we consider it legitimate to begin by exploring their points of view. A limitation of our study is that we collected data from single health centres in different provinces of the region and not from single local health care agencies that manage various health centres in a province. As described before, local health care agencies can influence the development of CP and CE processes in single health centres. However, very local-context, health centre-specific issues are also relevant; therefore, we decided to look at single good-practice examples.

The number of Community Health Centres that promote CP and CE is limited, and the examples of participatory processes to which the participants in the study referred, do not represent most approaches used in the Community Health Centres in Emilia-Romagna. However, due to the sample strategy employed, our results represent a large proportion of those health centres that support CP and CE, which is a strength of this sample. An investigation using a different sample (e. g., choosing local health care agencies as cases) would perhaps have drawn more attention to the problems and points of unclarity about these approaches. Nevertheless, considering the current developments of health centres in Italy, which have a mandate to foster

CP and CE, this research provides innovative examples for future discussions and implementation of CP and CE in Community Health Centres in general.

## Conclusions

Our study illustrates the ways in which CP and CE are defined by practitioners and stakeholders in terms of their implementation in practice and their aims in the context of Community Health Centres in Italy. The different understandings thus revealed both obstacles and opportunities that are valuable to practitioners, stakeholders, policymakers, and researchers. A joint understanding of CP and CE between the different involved actors such as practitioners, managers, community members and representatives, but also the local or regional policymakers, that takes into account the heterogeneity of communities and local contexts can be an important basis for action. A clear idea of who "the community" that should be involved is, should also consider the different needs of diverse community subgroups. This could imply, for instance, that in order to foster equity and representativeness when promoting CP and CE processes, involved actors should allocate enough time to talk about various group demands, explore common needs, and negotiate priorities. The varying ideas regarding the function of CP and CE highlight the need to reflect on the (full) potential of CP and CE beyond the level of health service development. As a result, policymakers and involved actors should take into account the various possible functions and benefits of CP and CE. In particular, they should reflect on the potential of a social justice perspective on CP, which is fundamental but often less tangible. To facilitate collaborative decision-making processes, practitioners and managers should consider the importance of addressing power disparities that exist between them and the community. This could mean discussing and eventually dealing with existing power imbalances or clarifying from the outset, the role and the amount of decision-making power that the community holds within specific processes. Finally, we highlighted the potential of CP and CE processes to support new professional attitudes. Policymakers as well as practitioners and managers could develop CP and CE approaches that actively engage practitioners so to enhance their comprehension, knowledge, and (experiential) expertise regarding the concepts of CP and CE. Further research should include the perspectives and experiences of community members and users of the Community Health Centres pertaining to CP and CE approaches to improve our understanding of how they perceive CP and CE, their aims, and the community's role in these approaches.

## Supporting information

**S1 Checklist. Inclusivity in global research.**
(DOCX)

**S1 File. Interview guideline practitioners and stakeholders.**
(PDF)

## Acknowledgments

The authors would like to thank the interview partners who gave their valuable time to participate in this study in such difficult times as COVID-19 pandemic.

## Author Contributions

**Conceptualization:** Daniela Rosalba Luisi, Kerstin Hämel.

**Data curation:** Daniela Rosalba Luisi, Kerstin Hämel.

**Formal analysis:** Daniela Rosalba Luisi, Kerstin Hämel.

**Investigation:** Daniela Rosalba Luisi.

**Methodology:** Daniela Rosalba Luisi, Kerstin Hämel.

**Project administration:** Daniela Rosalba Luisi.

**Supervision:** Kerstin Hämel.

**Writing – original draft:** Daniela Rosalba Luisi.

**Writing – review & editing:** Daniela Rosalba Luisi, Kerstin Hämel.

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
