## [Decision Letter · Decision Letter 0]

29 Apr 2024

PONE-D-23-42209Understandings of community participation and empowerment in primary health care in Emilia-Romagna, Italy: A qualitative interview study with practitioners and stakeholdersPLOS ONE

Dear Dr. Hämel,

Thank you for submitting your manuscript to PLOS ONE. After careful consideration, we feel that it has merit but does not fully meet PLOS ONE’s publication criteria as it currently stands. Therefore, we invite you to submit a revised version of the manuscript that addresses the points raised during the review process. The paper is really interesting, but some of the reviewers raised concerns about the lack of a theoretical framework. This is the more critical point of the reviews.Moreover, they required some clarifications about the selection of the sample that, in my opinion, are right. Therefore, I would ask you to improve the paper with their suggestions and to clarify some of the doubts that they had in reading your study.At the same time, I think that 19 interviews may be an adequate number and that the choice to select stakeholders within several different institutional levels can be relevant to the study. Please, you can find other comments below. 

We look forward to receiving your revised manuscript.

Kind regards,

Anna Prenestini, Ph.D.

Academic Editor

PLOS ONE

Journal Requirements:

3. In the online submission form, you indicated that given the potentially disclosive nature of entire interview transcripts they will not be made freely publicly available. They will be deposited at Bielefeld University and reasonable requests for secure research access will be considered. Please contact: kerstin.haemel@uni-bielefeld.de.

Additional Editor Comments:

The final responses of the three reviewers are completely different from each other.

Two reviewers raised crucial concerns about the lack of a comprehensive theoretical framework upon which the research is built.

Moreover, it seems that it is not completely clear which is the context of the sample. Have you considered only the stakeholders within only a Local Health Authority, or in a large area with several LHA, or the entire Emilia Romagna Region?

In the second or the latter case, are there any differences in the experts' responses due to the different contexts in the Emilia Romagna Region you have taken into account?

in my opinion, the paper is really interesting and useful for its results, but it requires major revisions. Please follow the previous and the other suggestions by the reviewers to improve your research paper.

Reviewers' comments:

Reviewer's Responses to Questions

**Comments to the Author**

1. Is the manuscript technically sound, and do the data support the conclusions?

Reviewer #1: Yes

Reviewer #2: Yes

Reviewer #3: Partly

2. Has the statistical analysis been performed appropriately and rigorously? 

Reviewer #1: No

Reviewer #2: N/A

Reviewer #3: Yes

3. Have the authors made all data underlying the findings in their manuscript fully available?

Reviewer #1: Yes

Reviewer #2: Yes

Reviewer #3: Yes

4. Is the manuscript presented in an intelligible fashion and written in standard English?

Reviewer #1: Yes

Reviewer #2: Yes

Reviewer #3: Yes

5. Review Comments to the Author

Reviewer #1: The topic is relevant and quite interesting indeed. the paper fails to present the literature review on this topic and specifically on that Emilia Romagna case. so it is not clear which is the debate the author(s) want to join and provide a contribution.

it is not clear if the 19 professionals are from the same LHA or not. It should because the LHA are very different and CdC are very different within the same Region (e.g. organized differently). The doctor are Gps or not ? if yes how they are selected?

It is almost impossible to benchmark two different LHA, so the analysis should have been done at regional level only or for a single LHA or otherwise using the same approch used by Longo et al or Compagni et al.

Reviewer #2: Dear authors,

I want to express my gratitude for allowing me to engage with your work titled "Understandings of community participation and empowerment in primary health care in Emilia-Romagna, Italy: A qualitative interview study with practitioners and stakeholders."

Your paper is well-written, concise, and engaging. It addresses several pertinent themes such as change management and community health, providing valuable insights. The abstract aptly encapsulates the essence of the article, and the research objective is clearly delineated. The methods employed are appropriate and well described. The results presented are significant, offering insights that practitioners, stakeholders, and policymakers can leverage, particularly regarding the conceptual categorization of community participation and empowerment within outpatient facilities like Case della Salute (CdS), now Case della Comunità (CdC). The discussion and conclusions effectively frame the results, highlighting both the accomplishments of the paper and its limitations.

While I believe the paper is well-suited for publication, I have some suggestions for revisions.

Title and Terminology

The paper lacks an explicit definition of primary care. It's worth considering that, according to both Italian and English terminologies, CdC extends beyond the boundaries of primary care. In many Beveridge systems, primary care typically refers to general practices offering gatekeeping services for adults and children. However, CdC aims to integrate various services and professional profiles, including specialist physicians, psychologists, obstetricians, etc. Therefore, I recommend revising the title and terminology throughout the paper to incorporate phrases such as "outpatient (healthcare) services" to better reflect the scope of CdC.

Results

While the themes regarding participants' understandings of CP and CE are compelling, there are a few aspects that could enhance the comprehensiveness and soundness of the results you present.

Firstly, I suggest indicating the exact number of interviewees who mentioned a specific theme, rather than relying on vague expressions like "According to many interview participants" (line 273).

Secondly, providing additional details about the professional profiles of the practitioners or stakeholders, such as whether they are nurses or physicians, would offer further context to the evidence presented.

Thirdly, the development of CP across CdC experiences appears to be uneven, influenced by various factors including the ideas and values of management and practitioners, as well as the availability of local resources (line 190). While this is a crucial issue, it is somewhat briefly addressed without providing direct quotations. I recommend expanding and elaborating on this aspect.

Best regards

Reviewer #3: Journal: Plos One

Manuscript ID: PONE-D-23-42209

Type: Research article

Title: Understandings of community participation and empowerment in primary health care in Emilia-Romagna, Italy: A qualitative interview study with practitioners and stakeholders

Synthesis: The study investigates the perceptions of community participation (CP) and empowerment (CE) among practitioners and stakeholders involved in primary health care (PHC) initiatives within the Emilia-Romagna region. The manuscript offers valuable insights into the varied understandings of CP and CE within the context of PHC, shedding light on the complexities inherent in participatory processes. The methodology employed, including semi-structured interviews and qualitative content analysis, appears appropriate for addressing the research objectives outlined in the abstract.

However, I believe that the manuscript requires significant revisions before it can be considered suitable for publication. My major concerns and suggestions for revision are outlined below:

INTRODUCTION:

The introduction of the paper outlines the significance of community participation (CP) and empowerment (CE) in primary healthcare (PHC) within the context of the Emilia-Romagna region in Italy.

To further enhance the introduction and clearly articulate the contribution to the international literature, consider expanding on the following points:

- The manuscript would benefit from a more explicit engagement with relevant theoretical frameworks or literature on community participation and empowerment. Incorporating theoretical perspectives could enrich the analysis and provide a stronger conceptual foundation for the study.

- Emphasize the unique characteristics of the Emilia-Romagna region that make it an important case study for understanding CP and CE in PHC. Highlight any distinctive features of the regional healthcare system.

By addressing these aspects, the introduction will provide a comprehensive overview of its contribution to the international literature on CP, CE, and PHC.

METHOD:

- I have some doubts about the size of the participants (N 19). The authors should explain more clearly the percentage of representativeness of the sample.

- Highlight the methodological rigor of your study, particularly in terms of data collection, analysis, and interpretation. Discuss how the use of qualitative content analysis and semi-structured interviews enhances the depth and richness of the findings, adding credibility to the study's contribution to the literature.

DISCUSSION:

Discuss how insights gained from studying CP and CE in Emilia-Romagna contribute to broader international discussions on PHC. Consider contrasting the findings with existing literature from other regions or countries to identify similarities, differences, and potential transferability of lessons learned.

6. PLOS authors have the option to publish the peer review history of their article (what does this mean?). If published, this will include your full peer review and any attached files.

Reviewer #1: No

Reviewer #2: **Yes: **Alberto Ricci

Reviewer #3: **Yes: **Monica Giancotti

---

## [Author Response · Author response to Decision Letter 0]

9 Jul 2024

We did a major revision and gave our best to address all the concerns and questions. We provide all the answers to the editor and the reviewers in the response-to-reviewers file.

---

## [Decision Letter · Decision Letter 1]

26 Aug 2024

Understandings of community participation and empowerment in primary health care in Emilia-Romagna, Italy: A qualitative interview study with practitioners and stakeholders

PONE-D-23-42209R1

Dear Dr. Hämel,

We’re pleased to inform you that your manuscript has been judged scientifically suitable for publication and will be formally accepted for publication once it meets all outstanding technical requirements.

Kind regards,

Anna Prenestini, Ph.D.

Academic Editor

PLOS ONE

Additional Editor Comments (optional):

The paper addressed each concern of the reviewers, so it is ready for publication in PLOS ONE.

Please refine the paper before the final upload.

Congratulations and kind regards,

Anna Prenestini, Academic Editor

Reviewers' comments:

Reviewer's Responses to Questions

**Comments to the Author**

1. If the authors have adequately addressed your comments raised in a previous round of review and you feel that this manuscript is now acceptable for publication, you may indicate that here to bypass the “Comments to the Author” section, enter your conflict of interest statement in the “Confidential to Editor” section, and submit your "Accept" recommendation.

Reviewer #2: All comments have been addressed

Reviewer #3: All comments have been addressed

2. Is the manuscript technically sound, and do the data support the conclusions?

Reviewer #2: Yes

Reviewer #3: Yes

3. Has the statistical analysis been performed appropriately and rigorously? 

Reviewer #2: N/A

Reviewer #3: Yes

4. Have the authors made all data underlying the findings in their manuscript fully available?

Reviewer #2: Yes

Reviewer #3: Yes

5. Is the manuscript presented in an intelligible fashion and written in standard English?

Reviewer #2: Yes

Reviewer #3: Yes

6. Review Comments to the Author

Reviewer #2: Thanks for allowing me to engage with the revision version of your work.

I think the authors have properly addressed all my issues/suggestions.

Best

Reviewer #3: Strengths:

1. Clarity and Depth of Analysis: The authors have provided a thorough and nuanced analysis of CP and CE, highlighting various themes and perspectives.

2. Methodological Rigor: The use of purposive and snowball sampling, as well as qualitative content analysis, is appropriate for the study's objectives. The triangulation of data sources and perspectives enhances the trustworthiness of the findings.

3. Contextual Relevance: The focus on Emilia-Romagna, a region with significant advancements in CP and CE, provides valuable insights that can inform policy and practice in similar contexts.

7. PLOS authors have the option to publish the peer review history of their article (what does this mean?). If published, this will include your full peer review and any attached files.

Reviewer #2: **Yes: **Alberto Ricci

Reviewer #3: **Yes: **Monica Giancotti

---

## [Editor Report · Acceptance letter]

17 Sep 2024

PONE-D-23-42209R1 

PLOS ONE

Dear Dr. Hämel, 

I'm pleased to inform you that your manuscript has been deemed suitable for publication in PLOS ONE. Congratulations! Your manuscript is now being handed over to our production team.

Kind regards, 

on behalf of

Professor Anna Prenestini 

Academic Editor

PLOS ONE